# B-Pref: Benchmarking Preference-Based Reinforcement Learning

**Kimin Lee, Laura Smith, Anca Dragan, Pieter Abbeel**
UC Berkeley

## Abstract

Reinforcement learning (RL) requires access to a reward function that incentivizes the right behavior, but these are notoriously hard to specify for complex tasks. Preference-based RL provides an alternative: learning policies using a teacher's preferences without pre-defined rewards, thus overcoming concerns associated with reward engineering. However, it is difficult to quantify the progress in preference-based RL due to the lack of a commonly adopted benchmark. In this paper, we introduce B-Pref: a benchmark specially designed for preference-based RL. A key challenge with such a benchmark is providing the ability to evaluate candidate algorithms quickly, which makes relying on real human input for evaluation prohibitive. At the same time, simulating human input as giving perfect preferences for the ground truth reward function is unrealistic. B-Pref alleviates this by simulating teachers with a wide array of irrationalities, and proposes metrics not solely for performance but also for robustness to these potential irrationalities. We showcase the utility of B-Pref by using it to analyze algorithmic design choices, such as selecting informative queries, for state-of-the-art preference-based RL algorithms. We hope that B-Pref can serve as a common starting point to study preference-based RL more systematically. Source code is available at https://github.com/rll-research/B-Pref.

## 1 Introduction

Deep reinforcement learning (RL) has emerged as a powerful method to solve a variety of sequential decision-making problems, including board games [58, 59], video games [10, 44, 68], autonomous control [9, 52], and robotic manipulation [5, 32, 35, 36]. However, scaling RL to many applications is difficult due to the challenges associated with defining a suitable reward function, which often requires substantial human effort. Specifying the reward function becomes harder as the tasks we want the agent to achieve become more complex (e.g., cooking or self-driving). In addition, RL agents are prone to exploit reward functions by discovering ways to achieve high returns in ways the reward designer did not expect nor intend. It is important to consider this phenomenon of reward exploitation, or reward hacking, since it may lead to unintended but dangerous consequences [28]. Further, there is nuance in how we might want agents to behave, such as obeying social norms that are difficult to account for and communicate effectively through an engineered reward function [4, 57, 67].

Preference-based RL [19, 31, 39] provides an alternative: a (human) teacher provides preferences between the two agent behaviors, and the agent then uses this feedback to learn desired behaviors (see Figure 1). This framework enables us to optimize the agent using RL without hand-engineered rewards by learning a reward function, which is consistent with the observed preferences [11, 12, 50]. Because a teacher can interactively guide agents according to their progress, preference-based RL has shown promising results (e.g., solving a range of RL benchmarks [19, 31], teaching novel behaviors [61, 71], and mitigating the effects of reward exploitation [39]).

35th Conference on Neural Information Processing Systems (NeurIPS 2021) Track on Datasets and Benchmarks.

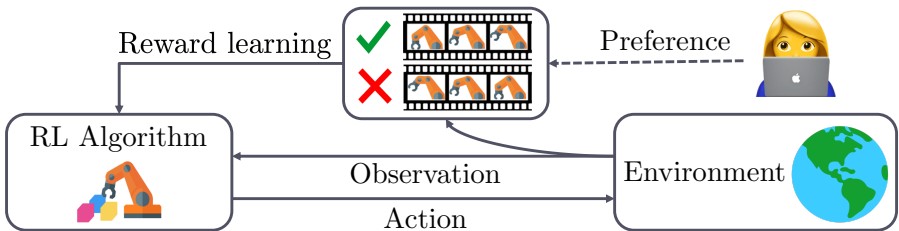

Figure 1: Illustration of preference-based RL. Instead of assuming that the environment provides a (hand-engineered) reward, a teacher provides preferences between the agent's behaviors, and the agent uses this feedback in order to learn the desired behavior.

Despite significant progress on RL benchmarks designed for various purposes (e.g., offline RL [24, 26], generalization [20, 21], meta RL [76], and safe RL [48]), existing benchmarks are not tailored towards preference-based RL. The lack of a standard evaluation benchmark makes it hard to quantify scientific progress. Indeed, without consistent evaluation, it is not easy to understand the effects of algorithmic and design decisions or compare them across papers.

In this paper, we introduce B-Pref: a benchmark for preference-based RL consisting of various loco-motion and robotic manipulation tasks from DeepMind Control Suite [65, 66] and Meta-world [76]. While utilizing real human input is ideal, this is prohibitive because it is hard to evaluate candidate algorithms quickly using real human input. Prior works [19, 31, 39] address this issue by simulating human input as giving perfect preferences with respect to an underlying ground truth reward function. However, evaluation on such ideal teachers is unrealistic because actual humans can exhibit various irrationalities [18] in decision making. So, in our benchmark, we design simulated human teachers with a wide array of irrationalities and propose evaluation metrics not solely for performance but also for robustness to these potential irrationalities.

To serve as a reference, we benchmark state-of-the-art preference-based RL algorithms [19, 39] in B-Pref and showcase the utility of B-Pref by using it to analyze algorithmic design choices for preference-based RL. Although existing methods provide fairly efficient performance on perfectly rational teachers, the poor performance on more realistic, irrational teachers calls for new algorithms to be developed.

The benchmark and reference implementations are available at `https://github.com/rll-research/B-Pref`. We believe that systematic evaluation and comparison will not only further our understanding of the strengths of existing algorithms, but also reveal their limitations and suggest directions for future research.

## 2 Preliminaries

We consider an agent interacting with an environment in discrete time [63]. At each timestep $t$, the agent receives a state $\mathbf{s}_t$ from the environment and chooses an action $\mathbf{a}_t$ based on its policy $\pi$.

In traditional reinforcement learning, the environment also returns a reward $r(\mathbf{s}_t, \mathbf{a}_t)$ and the goal of agent is to maximize the discounted sum of rewards. However, for many complex domains and tasks, it is difficult to construct a suitable reward function. We consider the preference-based RL framework, where a (human) teacher provides preferences between the agent's behaviors and the agent uses this feedback to perform the task [19, 31, 39, 41].

Formally, a segment $\sigma$ is a sequence of observations and actions $\{(\mathbf{s}_1, \mathbf{a}_1), ..., (\mathbf{s}_H, \mathbf{a}_H)\}$. Given a pair of segments $(\sigma^0, \sigma^1)$, a teacher indicates which segment is preferred, i.e., $y = (0, 1)$ or $(1, 0)$, that the two segments are equally preferred $y = (0.5, 0.5)$, or that two segments are incomparable, i.e., discarding the query. The goal of preference-based RL is to train an agent to perform behaviors desirable to a human teacher using as few queries as possible.

**Algorithm 1** SimTeacher: Simulated human teachers
___
**Require:** Discount factor $\gamma$, rationality constant $\beta$, probability of making a mistake $\epsilon$
**Require:** Skip threshold $\delta_{\texttt{skip}}$, equal threshold $\delta_{\texttt{equal}}$
**Require:** Pair of segments $\sigma^0, \sigma^1$
1: **if** $\max_{i \in \{0,1\}} \sum_t r\left(\mathbf{s}_t^i, \mathbf{a}_t^i\right) < \delta_{\texttt{skip}}$ **then** // SKIPPING QUERY
2:      $y \leftarrow \emptyset$
3: **else if** $\left| \sum_t r\left(\mathbf{s}_t^1, \mathbf{a}_t^1\right) - \sum_t r\left(\mathbf{s}_t^0, \mathbf{a}_t^0\right) \right| < \delta_{\texttt{equal}}$ **then** // EQUALLY PREFERABLE
4:      $y \leftarrow (0.5, 0.5)$
5: **else if** $\sigma^0 \succ \sigma^1 \sim P[\sigma^0 \succ \sigma^1; \beta, \gamma]$ **then** // SAMPLING PREFERENCES FROM (1)
6:      $y \leftarrow (1, 0)$ with probability of $1 - \epsilon$
7:      $y \leftarrow (0, 1)$ otherwise // MAKING A MISTAKE
8: **else**
9:      $y \leftarrow (0, 1)$ with probability of $1 - \epsilon$
10:      $y \leftarrow (1, 0)$ otherwise // MAKING A MISTAKE
11: **end if**
12: **return** $y$
___

## 3    B-Pref: Benchmarks environments for preference-based RL

### 3.1    Design factors

While ideally we would evaluate algorithms' real-world efficacy using real human feedback, designing a standardized and broadly available benchmark becomes challenging because we do not have ground truth access to the human's reward function. Instead, we focus on solving a range of existing RL tasks (see Section 3.4) using a simulated human, whose preferences are based on a ground truth reward function. Because simulated human preferences are immediately generated by the ground truth reward, we are able to evaluate the agent quantitatively by measuring the true average return and do more rapid experiments. A major challenge with simulating human input is that real humans are not perfectly rational and will not provide perfect preferences. To alleviate this challenge, we propose to simulate human input using a wide array of irrationalities (see Section 3.2), and measure an algorithm's robustness in handling such input (see Section 3.3).

### 3.2    Simulated human teachers

We start from a (perfectly rational) deterministic teacher, which generates preferences as follows:

$$y = \begin{cases} (1,0) & \text{If } \sum_{t=1}^H r(\mathbf{s}_t^0, \mathbf{a}_t^0) > \sum_{t=1}^H r(\mathbf{s}_t^1, \mathbf{a}_t^1) \\ (0,1) & \text{otherwise,} \end{cases}$$

where $H > 0$ is a length of segment $\sigma$ and $r$ is the ground truth reward. We remark that prior works [19, 31, 39] evaluated their methods using this ideal teacher. However, evaluating the performance of preference-based RL only using the ideal teacher is unrealistic because there are many possible irrationalities [17, 18] affecting a teacher's preferences (and expression of preferences) in different ways.

To design more realistic models of human teachers, we consider a common stochastic model [11, 19, 50] and systematically manipulate its terms and operators (see Algorithm 1):

**Stochastic preference model.** Because preferences from the human can be noisy, we generate preferences using a stochastic model defined as follows (Line 5):

$$P[\sigma^i \succ \sigma^j; \beta, \gamma] = \frac{\exp\left(\beta \sum_{t=1}^H \gamma^{H-t} r(\mathbf{s}_t^i, \mathbf{a}_t^i)\right)}{\exp\left(\beta \sum_{t=1}^H \gamma^{H-t} r(\mathbf{s}_t^i, \mathbf{a}_t^i)\right) + \exp\left(\beta \sum_{t=1}^H \gamma^{H-t} r(\mathbf{s}_t^j, \mathbf{a}_t^j)\right)}, \quad (1)$$

where $\gamma \in (0, 1]$ is a discount factor to model myopic behavior, $\beta$ is a rationality constant, and $\sigma^i \succ \sigma^j$ denotes the event that segment $i$ is preferable to segment $j$. This follows the Bradley-Terry model [13], which can be interpreted as assuming the probability of preferring a segment depends exponentially on the sum over the segment of an underlying reward. Note that this teacher becomes a perfectly rational and deterministic as $\beta \to \infty$, whereas $\beta = 0$ produces uniformly random choices.

**Myopic behavior**. Humans are sometimes myopic (short-sighted), so a human teacher may remember and focus more on the behavior at the end of the clip they watched, for example. We model myopic behavior by introducing a weighted sum of rewards with a discount factor $\gamma$ in (1), i.e., $\sum_{t=1}^{H} \gamma^{H-t} r(\mathbf{s}_t^i, \mathbf{a}_t^i)$, so that our simulated teacher places more weight on recent timesteps.

**Skipping queries**. If both segments do not contain a desired behavior, a teacher would like to mark them as incomparable and discard the query. We model this behavior by skipping a query if the sum over the segment of an underlying reward is smaller than skip threshold $\delta_{\texttt{skip}}$, i.e., $\max_{i \in \{0,1\}} \sum_t r\left(\mathbf{s}_t^i, \mathbf{a}_t^i\right) < \delta_{\texttt{skip}}$ (Line 1).

**Equally preferable**. If the two segments are equally good, instead of selecting one segment as preferable, a teacher would like to mark the segments as equally preferable. Motivated by this, we provide an uniform distribution $(0.5, 0.5)$ as a response (Line 3) if both segments have similar sum of rewards, i.e., $\left| \sum_t r\left(\mathbf{s}_t^1, \mathbf{a}_t^1\right) - \sum_t r\left(\mathbf{s}_t^0, \mathbf{a}_t^0\right) \right| < \delta_{\texttt{equal}}$.

**Making a mistake**. Humans can make accidental errors when they respond. To reflect this, we flip the preference with probability of $\epsilon$ (Line 7 and Line 10).

### 3.3 Evaluation metrics

We evaluate two key properties of preference-based RL: *performance of the RL agent under a fixed budget of feedback* and *robustness to potential irrationalities*. Because the simulated human teacher's preferences are generated by a ground truth reward, we measure the true average return of trained agents as evaluation metric. To facilitate comparison across different RL algorithms, we normalize returns with respect to the baseline of RL training using the ground truth reward:

$$\texttt{Normalized returns} = \frac{\texttt{Average returns of preference-based RL}}{\texttt{Average returns of RL with ground truth reward}}.$$

To evaluate the feedback-efficiency of preference-based RL algorithms, we compare normalized returns by varying the maximum budget of queries.

To evaluate the robustness, we evaluate against the following simulated human teachers with different properties:

- Oracle: $\texttt{SimTeacher}\left( \beta \to \infty, \gamma = 1, \epsilon = 0, \delta_{\texttt{skip}} = 0, \delta_{\texttt{equal}} = 0 \right)$
- Stoc: $\texttt{SimTeacher}\left( \beta = 1, \gamma = 1, \epsilon = 0, \delta_{\texttt{skip}} = 0, \delta_{\texttt{equal}} = 0 \right)$
- Mistake: $\texttt{SimTeacher}\left( \beta \to \infty, \gamma = 1, \epsilon = 0.1, \delta_{\texttt{skip}} = 0, \delta_{\texttt{equal}} = 0 \right)$
- Skip: $\texttt{SimTeacher}\left( \beta \to \infty, \gamma = 1, \epsilon = 0, \delta_{\texttt{skip}} > 0, \delta_{\texttt{equal}} = 0 \right)$
- Equal: $\texttt{SimTeacher}\left( \beta \to \infty, \gamma = 1, \epsilon = 0, \delta_{\texttt{skip}} = 0, \delta_{\texttt{equal}} > 0 \right)$
- Myopic: $\texttt{SimTeacher}\left( \beta \to \infty, \gamma = 0.9, \epsilon = 0, \delta_{\texttt{skip}} = 0, \delta_{\texttt{equal}} = 0 \right)$

In our evaluations, we consider one modification (i.e., irrationality) to the oracle teacher at a time, which allows us to isolate the individual effects. While each individually may not exactly model real human behavior, it would be straightforward to use our benchmark to create more complex teachers that combine multiple irrationalities.

### 3.4 Tasks

We consider two locomotion tasks (Walker-walk and Quadruped-walk) from DeepMind Control Suite (DMControl) [65, 66] and two robotic manipulation tasks (Button Press and Sweep Into) from Meta-world [76]. We focus on learning from the proprioceptive inputs and dense rewards because learning from visual observations and sparse rewards can cause additional issues, such as representation learning [37, 55, 60, 62, 74] and exploration [56]. However, we think it is an interesting and important direction for future work to consider visual observations and sparse rewards.

# 4 B-Pref: Algorithmic baselines for preference-based RL

Throughout this paper, we mainly focus on two of the most prominent preference-based RL algorithms [19, 39], which involve reward learning from preferences. Formally, a policy $\pi_\phi$ and reward function $\widehat{r}_\psi$ are updated as follows (see Algorithm 3 in the supplementary material):

- *Step 1 (agent learning)*: The policy $\pi_\phi$ interacts with environment to collect experiences and we update it using existing RL algorithms to maximize the sum of the learned rewards $\widehat{r}_\psi$.
- *Step 2 (reward learning)*: We optimize the reward function $\widehat{r}_\psi$ via supervised learning based on the feedback received from a teacher.
- Repeat *Step 1* and *Step 2*.

## 4.1 Deep reinforcement learning from human preferences

In order to incorporate human preferences into deep RL, Christiano et al. [19] proposed a framework that learns a reward function $\widehat{r}_\psi$ from preferences [50, 70]. Specifically, we first model a preference predictor using the reward function $\widehat{r}_\psi$ as follows:

$$P_\psi[\sigma^1 \succ \sigma^0] = \frac{\exp \sum_t \widehat{r}_\psi(\mathbf{s}_t^1, \mathbf{a}_t^1)}{\sum_{i \in \{0,1\}} \exp \sum_t \widehat{r}_\psi(\mathbf{s}_t^i, \mathbf{a}_t^i)}, \tag{2}$$

where $\sigma^i \succ \sigma^j$ denotes the event that segment $i$ is preferable to segment $j$. We remark that this corresponds to assume a stochastic teacher following the Bradley-Terry model [13] but we do not assume that the type and degree of irrationality or systematic bias is available in our experiments. Because this could lead to a poor preference inference [17], future work may be able to further improve the efficiency of learning by approximating the teacher's irrationality.

To align our preference predictor with the teacher's preferences, we consider a binary classification problem using the cross-entropy loss. Specifically, given a dataset of preferences $\mathcal{D}$, the reward function, modeled as a neural network with parameters $\psi$, is updated by minimizing the following loss:

$$\mathcal{L}^{\texttt{Reward}} = - \mathop{\mathbb{E}}_{(\sigma^0, \sigma^1, y) \sim \mathcal{D}} \left[ y(0) \log P_\psi[\sigma^0 \succ \sigma^1] + y(1) \log P_\psi[\sigma^1 \succ \sigma^0] \right]. \tag{3}$$

Once we learn a reward function $\widehat{r}_\psi$, we can update the policy $\pi_\phi$ using any RL algorithm. A caveat is that the reward function may be non-stationary because we update it during training. To mitigate the effects of a non-stationary reward function, Christiano et al. [19] used on-policy RL algorithms, such as TRPO [52] and A2C [45]. We re-implemented this method using the state-of-the-art on-policy RL algorithm: PPO [54]. We refer to this baseline as PrefPPO.

## 4.2 PEBBLE

PEBBLE [39] is a state-of-the-art preference-based RL algorithm that improved the framework of Christiano et al. [19] using the following ideas:

**Unsupervised pre-training**. In the beginning of training, a naive agent executing a random policy does not provide good state coverage nor coherent behaviors. Therefore, the agent's queries are not diverse and a teacher can not convey much meaningful information. As a result, it requires many samples (and thus queries) for these methods to show initial progress. Ibarz et al. [31] has addressed this issue by assuming that demonstrations are available at the beginning of the experiment. However, this is not ideal since suitable demonstrations are often prohibitively expensive to obtain in practice. Instead, PEBBLE pre-trains the policy only using intrinsic motivation [46, 51] to learn how to generate diverse behaviors. Specifically, by updating the agent to maximize the state entropy $\mathcal{H}(\mathbf{s}) = -\mathbb{E}_{\mathbf{s} \sim p(\mathbf{s})} [\log p(\mathbf{s})]$, it encourages the agent to efficiently explore an environment and collect diverse experiences (see the supplementary material for more details).

**Off-policy RL with relabeling**. To overcome the poor sample-efficiency of on-policy RL algorithms, PEBBLE used the state-of-the-art off-policy RL algorithm: SAC [27]. However, the learning process can be unstable because previous experiences in the replay buffer are labeled with previous learned rewards. PEBBLE stabilizes the learning process by relabeling all of the agent's past experience every time it updates the reward model.

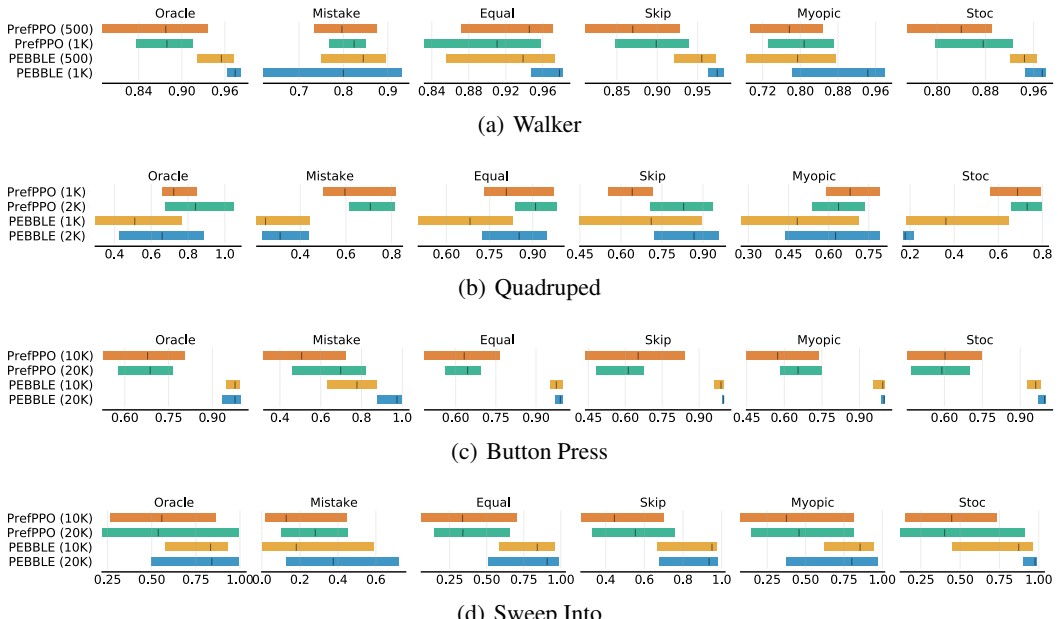

(a) Walker

(b) Quadruped

(c) Button Press

(d) Sweep Into

Figure 2: IQM normalized returns with 95% confidence intervals across ten runs. Learning curves and other metrics (median, mean, optimality gap) are in the supplementary material.

## 5 Using B-Pref to analyze algorithmic design decisions

We design our experiments to investigate the following:

- How do existing preference-based RL methods compare against each other across environments with different complexity?
- How to use B-Pref to analyze algorithmic design decisions for preference-based RL?

### 5.1 Training details

We implement PEBBLE and PrefPPO using publicly released implementations of SAC[1] and PPO.[2] All hyperparameters of all algorithms are optimized independently for each environment. All of the experiments were processed using a single GPU (NVIDIA GTX 1080 Ti) and 8 CPU cores (Intel Xeon Gold 6126). For reliable evaluation [1], we measure the normalized returns[3] and report the interquartile mean (IQM) across ten runs using an open-source library *rliable*.[4] More experimental details (e.g., model architectures and the final hyperparameters) and all learning curves with standard deviation are in the supplementary material.

### 5.2 Benchmarking prior methods

Figure 2 shows the IQM normalized returns of PEBBLE and PrefPPO at convergence on various simulated teachers listed in Section 3.2 (see the supplementary material for experimental details). For a fair comparison, we apply unsupervised pre-training and disagreement-based sampling to all methods (including SAC and PPO). PEBBLE outperforms PrefPPO in most of the environments (especially achieving large gains on robotic manipulation tasks). Interestingly, providing uniform labels to equally preferable segments (Equal) or skipping the queries with similar behaviors (Skip) is more useful than relying only on perfect labels (Oracle) on hard environments like Quadruped (Figure 2(b)). While both PEBBLE and PrefPPO achieve fairly efficient performance on correct

---

[1]https://github.com/denisyarats/pytorch_sac
[2]https://github.com/DLR-RM/stable-baselines3
[3]On robotic manipulation tasks, we measure the task success rate as defined by the Meta-world authors [76].
[4]https://github.com/google-research/rliable

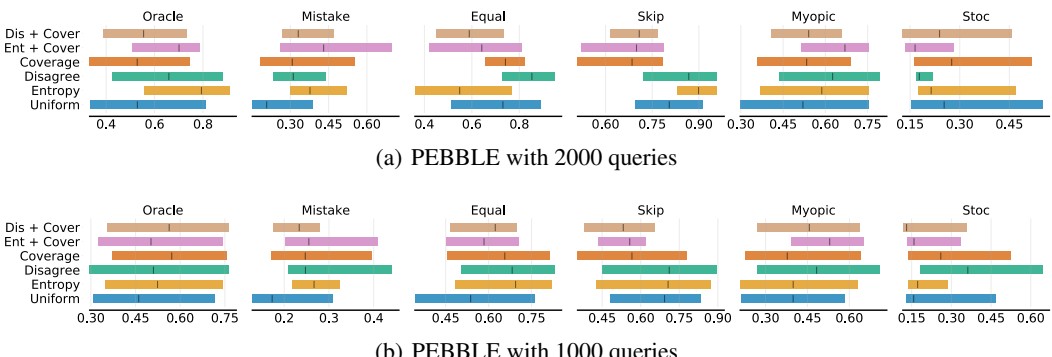

(a) PEBBLE with 2000 queries

(b) PEBBLE with 1000 queries

Figure 3: IQM normalized returns of PEBBLE with various sampling schemes across ten runs on Quadruped. Learning curves and other metrics (median, mean, optimality gap) are in the supplementary material.

labels (Oracle, Equal and Skip), they often suffer from poor performance when teachers can provide the wrong labels (Mistake and Stoc). This suggests opportunities for further investigations and development of techniques that can improve the robustness to corrupted labels.[5]

## 5.3 Impact of design decisions in reward learning

Reward learning from preferences involves several design decisions, which can affect the performance of the overall framework. We showcase the utility of B-Pref by using it to analyze the following algorithmic design choices in depth:

**Selecting informative queries**. During training, all experiences are stored in an annotation buffer $\mathcal{B}$ and we generate $N_{\texttt{query}}$ pairs of segments[6] to ask teacher's preferences from this buffer at each feedback session. To reduce the burden on the human, we should solicit preferences so as to maximize the information received. While finding optimal queries is computationally intractable [2], several sampling schemes [11, 12, 50] have been explored to find queries that are likely to change the reward model. Specifically, we consider the following sampling schemes, where more details are in the supplementary material:

- *Uniform sampling*: We pick $N_{\texttt{query}}$ pairs of segments uniformly at random from the buffer $\mathcal{B}$.
- *Uncertainty-based sampling*: We first generate the initial batch of $N_{\texttt{init}}$ pairs of segments $\mathcal{G}_{\texttt{init}}$ uniformly at random, measure the uncertainty (e.g., variance across ensemble of preference predictors [19] or entropy of a single preference predictor [39]), and then select the $N_{\texttt{query}}$ pairs of segments with high uncertainty.
- *Coverage-based sampling*: From the initial batch $\mathcal{G}_{\texttt{init}}$, we choose $N_{\texttt{query}}$ center points such that the largest distance between a data point and its nearest center is minimized using a greedy selection strategy.
- *Hybrid sampling*: Similar to Yu et al. [75], we also consider hybrid sampling, which combines uncertainty-based sampling and coverage-based sampling. First, we select the $N_{\texttt{inter}}$ pairs of segments $\mathcal{G}_{\texttt{un}}$ using uncertainty-based sampling, where $N_{\texttt{init}} > N_{\texttt{inter}}$, and then choose $N_{\texttt{query}}$ center points from $\mathcal{G}_{\texttt{un}}$.

Figure 3 shows the IQM normalized returns of PEBBLE with various sampling schemes on Quadruped. We find that the uncertainty-based sampling schemes (i.e., ensemble disagreement and entropy) are superior to other sampling schemes, while coverage-based sampling schemes do not improve on uniform sampling and slow down the sampling procedures. To analyze the effects of sampling schemes, we measure the fraction of equally preferable queries (i.e., $y = (0.5, 0.5)$) on

---

[5]We find that label smoothing [64] is not effective in handling corrupted labels in our experiments (see the supplementary material for supporting results). However, other regularization techniques, like label flipping, L2 regularization and weight decay, would be interesting for further study in future work.

[6]We do not compare segments of different lengths.

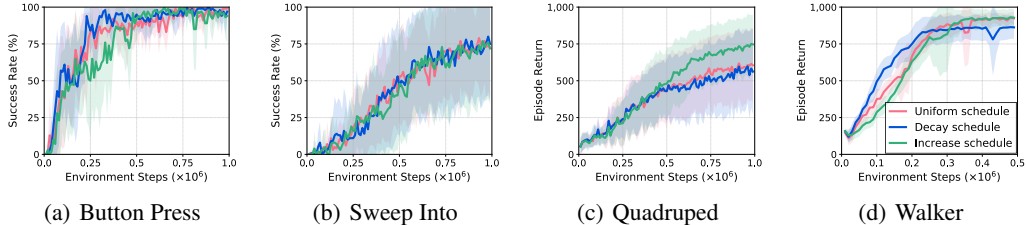

(a) Button Press      (b) Sweep Into      (c) Quadruped      (d) Walker

Figure 4: Learning curves of PEBBLE with different feedback schedules on the oracle teacher. The solid line and shaded regions represent the mean and standard deviation, respectively, across ten runs.

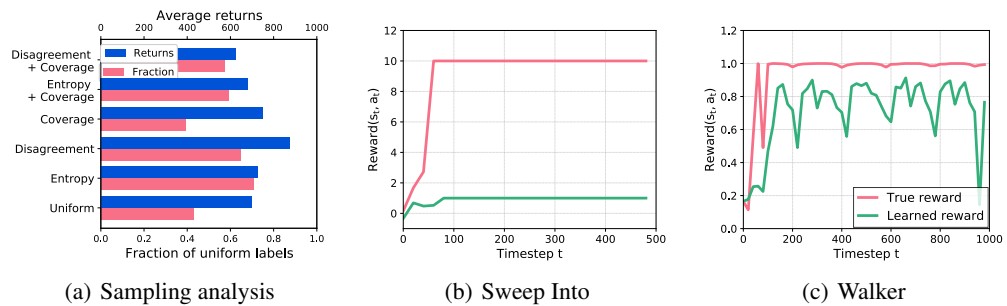

(a) Sampling analysis      (b) Sweep Into      (c) Walker

Figure 5: (a) Fraction of equally preferable queries (red) and average returns (blue) on the Equal teacher. We use PEBBLE with different sampling schemes on Quadruped given a budget of 2000 queries. Even though a teacher provides more uniform labels, i.e., $y = (0.5, 0.5)$, to uncertainty-based sampling schemes, they achieve higher returns than other sampling schemes. (b/c) Time series of learned reward function (green) and the ground truth reward (red) using rollouts from a policy optimized by PEBBLE. Learned reward functions align with the ground truth rewards in (b) Sweep Into and (c) Walker.

the Equal teacher. Figure 5(a) shows that uncertainty-based sampling schemes achieve high returns even though other sampling schemes receive more (non-uniform) perfect labels. We expect that this is because queries with high uncertainty provide significant information to the reward model. This also suggests opportunities for further investigations on uncertainty estimates like Bayesian methods [15, 25].

**Feedback schedule**. We also investigate the impact of the feedback schedule, which decides the number of queries at each feedback session. Lee et al. [39] used a uniform schedule, which always asks the same number of queries, and Christiano et al. [19], Ibarz et al. [31] used a decay schedule, which decreases the number of queries, roughly proportional to $\frac{T}{t+T}$, where $t$ is the current timestep and $T$ is the episode length. We additionally consider an increase schedule, which increases the number of queries, roughly proportional to $\frac{T+t}{T}$.

Figure 4 shows the learning curves of PEBBLE with different feedback schedule on the oracle teacher. Given the same total number of queries, increase and decay schedules change the size of the initial queries by a factor of 0.5 and 2, respectively. One can note that there is no big gain from rule-based schedules in most of the environments. Even though rule-based schedules are less effective than uniform scheduling, using an adaptive schedule like meta-gradient [72, 73] would be interesting for further study in future work.

**Reward analysis**. To investigate the quality of the learned reward function, we compare the learned reward function with the ground truth reward. Figure 5(b) and Figure 5(c) show the learned reward function optimized by PEBBLE on the oracle teacher in Sweep Into and Walker, where more evaluation results on other environments are also available in the supplementary material. Because we bound the output of the reward function using tanh function, the scale is different with the ground truth reward but the learned reward function is reasonably well-aligned.

# 6 Related work

**Benchmarks for deep reinforcement learning**. There is a large body of work focused on designing benchmarks for RL [7, 8, 14, 20–22, 24, 26, 30, 48, 65, 66, 76]. The Arcade Learning Environment [8] has becomes a popular benchmark to measure the progress of RL algorithms for discrete control tasks. For continuous control tasks, Duan et al. [22] presented a benchmark with baseline implementations of various RL algorithms, which in turn led to OpenAI Gym [14]. These benchmarks have significantly accelerated progress and have been strong contributors towards the discovery and evaluation of today's most widely used RL algorithms [27, 44, 52–54].

Recently, researchers proposed more targeted RL benchmarks that have been designed for specific research purposes. Cobbe et al. [21] presented a suite of game-like environments where the train and test environments differ for evaluating generalization performance of RL agents. Ray et al. [48] provided a Safety Gym for measuring progress towards RL agents that satisfy the safety constraints. D4RL [24] and RL Unplugged [26] have been proposed to evaluate and compare offline RL algorithms. Yu et al. [76] proposed Meta-world to study meta- and multi-task RL. URLB [38] benchmarks performance of unsupervised RL methods. However, none of the existing RL benchmarks are tailored towards preference-based RL.

Freire et al. [23] proposed DERAIL, a benchmark suite for preference-based learning, but they focused on simple diagnostic tasks. In B-Pref, we consider learning a variety of complex locomotion and robotic manipulation tasks. Additionally, we design teachers with a wide array of irrationalities and benchmark state-of-the-art preference-based RL algorithms [19, 39] in depth.

**Human-in-the-loop reinforcement learning**. Several works have successfully utilized feedback from real humans to train RL agents [6, 19, 31, 34, 39, 43, 69]. MacGlashan et al. [43] proposed a reward-free method, which utilizes a human feedback as an advantage function and optimizes the agents via a policy gradient. Knox & Stone [34] trained a reward model via regression using unbounded real-valued feedback. However, these approaches are difficult to scale to more complex learning problems that require substantial agent experience.

Another promising direction has focused on utilizing the human preferences [3, 19, 31, 39, 41, 47, 61, 70, 71]. Christiano et al. [19] scaled preference-based learning to utilize modern deep learning techniques, and Ibarz et al. [31] improved the efficiency of this method by introducing additional forms of feedback such as demonstrations. Recently, Lee et al. [39] proposed a feedback-efficient RL algorithm by utilizing off-policy learning and pre-training. Stiennon et al. [61] and Wu et al. [71] showed that preference-based RL can be utilized to fine-tune GPT-3 [16] for hard tasks like text and book summarization, respectively. We benchmark these state-of-the-art preference-based RL algorithms in this paper.

# 7 Conclusion

In this paper, we present B-Pref, a benchmark specially designed for preference-based RL, covering a wide array of a teacher's irrationalities. We empirically investigate state-of-the-art preference-based RL algorithms in depth and analyze the effects of algorithmic design decisions on our benchmark. We find that existing methods often suffer from poor performance when teachers provide wrong labels, and the effects of design decisions are varied depending on the task setups. These observations call for new algorithms in active learning [11, 12, 50] and meta-learning [72, 73] to be developed. By providing an open-source release of the benchmark, we encourage other researchers to use B-Pref as a common starting point to study preference-based RL more systematically.

**Limitations.** There are several important properties that are not explored in-depth in B-Pref. One is robustness of learned reward functions to new environments with different dynamics or initial states [49]. Also, we focus on tasks with proprioceptive inputs and dense rewards, but extensions to visual observations and sparse rewards are interesting directions to explore.

**Potential negative impacts.** Preference-based RL has several advantages (e.g., teaching novel behaviors, and mitigating the effects of reward exploitation); however, it also has potential drawbacks. Malicious users might teach the bad behaviors/functionality using this framework. Therefore, researchers should consider the safety issues with particular thought.

## Acknowledgements

This research is supported in part by ONR PECASE N000141612723, NSF NRI #2024675, ONR YIP, and Berkeley Deep Drive. Laura Smith was supported by NSF Graduate Research Fellowship. We thank Qiyang (Colin) Li and Olivia Watkins for providing helpful feedback and suggestions. We also thank anonymous reviewers for critically reading the manuscript and suggesting substantial improvements.

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
