# OpenReview forum: "B-Pref: Benchmarking Preference-Based Reinforcement Learning"
_NeurIPS.cc/2021/Track/Datasets_and_Benchmarks/Round1 — NeurIPS 2021 Datasets and Benchmarks Track (Round 1)_

### Official Review · Reviewer_chBv · 2021-07-03
**A new benchmark for preference based reinforcement learning**

**Rating:** 8
**Confidence:** 3
**Clarity:** Yes. I enjoyed reading it.

**Strengths:**

(1) Paper is well written and easy to understand.
(2) I expect this work to be very influential for preference based RL research. Should provide researchers the avenue to work on algorithmic design and benchmark their algorithms against the current state of the art very quickly.
(3) Code seems easy to run and install using pip. This is very important - often the largest barrier in the proliferation of such works is how much time it can take to set things up. The installation page is very simple and it seems very easy to run experiments.

**Weaknesses:**

(1) It would be interesting if any agents could be trained using actual human feedback. This is not a priority but an interesting follow up would be to actually quantify some of the "irrationalities" and mistakes that humans make.
(2) A laymans introduction / paragraph in the paper describing how preference based RL is used would be beneficial. It would be nice if the technical definition from section 2 could be juxtaposed with a simple example that highlights two segments of an agent learning a behavior like a "mujoco robot trying to learn a backflip".

**Additional Feedback:**

None

**Correctness:**

I believe so. Evaluation and experimentation are easy to understand and appear rational.

**Documentation:**

Yes.

**Ethics:**

No.

**Relation To Prior Work:**

Related work talks about prior benchmarking approaches in RL. While preference based RL is of recent interest, a lack of benchmarking makes progress hard to measure. The paper appears to be well placed and quite different from the cited works.

**Summary And Contributions:**

Summary:
Preference based RL is of active interest: utilizing human preferences to train agents allows to creation of agents that whose motivations are better understood. Preference based RL can be easier than expecting a reward function that can be very complicated to create and relies on expert human knowledge of the task for its formulation. The authors introduce B-Pref, a benchmark specifically designed for testing preference-based RL approaches. Humans are simulated based on ground truth data. Since humans are not perfectly rational beings, they are also simulated with several "irrationalities" to benchmark algorithms according to more realistic scenarios.

Contributions:
(1) B-Pref - a new benchmark for preference-based RL
(2) Simulating human decisions including examples of irrationality
(3) Benchmarking of current SOTA RL algorithms on B-pref
(4) Showcasing that these algorithms often do not perform very well when the "humans" provide noisy / mistaken data and identifying a key direction in which research in this field can progress. \
(5) Easily understandable analysis of how to sample queries so as to maximize learning when the humans must provide decisions.

---

> ### Author Response · Authors · 2021-07-10
> **Authors' Response**
>
> We sincerely thank you for your helpful feedback and insightful comments. We appreciate that our paper is recognized for several positive aspects: (1) New benchmark with various irrational teachers and well-designed evaluation protocol, (2) benchmarking current SOTA preference-based RL algorithms, (3) detailed analysis on different algorithmic design choices, (4) clear write-up, and (5) well-documented (easy to install and run) codebase. We address your comments and questions below:
>
> ---
> **Q1. Real human teacher**
>
> **A1.** We agree that showing a relationship between the irrationality models and real human teaching would be very valuable, however, the reality is that human behavior is nuanced and complex. Even though we picked representative biases (see A2 of Reviewer zxDS; https://openreview.net/forum?id=ps95-mkHF_&noteId=KYlCUFmG0re), chances are that none of the biases in our simulated teachers perfectly correlate with real human behavior (rather, human behavior is explained by some combination of some aspects of these with others that we do not yet model in the benchmark). The goal of the irrationalities is not to be a perfect model of human behavior, rather to enable evaluating the method with a diversity of possible teachers. The idea is that if a method can perform well with multiple teachers in the benchmark, then it has stronger chances of performing well with real teachers as well. In the future, we plan a follow-up user study that looks not so much at the correlation between the simulated teachers and real ones, but rather the correlation between the reward learning performance as evaluated by the benchmark, and the performance as evaluated with real humans. We believe that positive results there would serve to further strengthen the value of the benchmark.
>
> ---
> **Q2. Editorial comments**
>
> **A2.** Thank you for the suggestion to improve the writing. We will incorporate the suggested editorial comments in the final draft.

---

### Official Review · Reviewer_V29j · 2021-07-04
**Review for B-Pref: Benchmarking Preference-Based Reinforcement Learning**

**Rating:** 7
**Confidence:** 3
**Correctness:** No obvious incorrectness is noticed.
**Clarity:** The paper is well written.

**Strengths:**

- systematic modeling of human noise and irretionality for giving the preference;
- use the benchmarking to investigate the effect of different algorithmic design choices including sampling strategy and feedback schedule, which could shed light on future algorithm design

**Weaknesses:**

- the scale of the benchmarked environments is a bit small, i.e., only 4 environments; It would be interesting to see if the conclusion would hold the same on more environments.


**Additional Feedback:**

I understand it could be expensive to collect real human feedbacks, but can the author provide some more justifications (e.g., refering to previous works on human behaviour study) on why the irrationality model proposed in the paper is accurate enough?

**Documentation:**

The benchmarking is well documented.

**Ethics:**

No; the paper already has a discussion on considering saftty issues to avoid malicious users  to teach bad behaviours.

**Relation To Prior Work:**

The difference from previous contributions are well discussed.

**Summary And Contributions:**

This paper proposes a new benchmarking for preference-based RL. It introduces several techniques to model the irrationality and noise of humans for giving the preference such as myopic behaviour, skipping queires, and etc. It benchmarked two state-of-the-art preference-based RL algorithms on 4 locomtion and robotics manipulation tasks with variants of irrational models and different budget of queries, and gave detailed analysis of the efficiency and robustness of these methods. It also used the benchmarking to study the impact of different design choices in learning rewards from preferences. This paper could have large potential in boosting the algorithm development for preference-based RL.

---

> ### Author Response · Authors · 2021-07-10
> **Authors' Response**
>
> We sincerely thank you for your helpful feedback and insightful comments. We appreciate that our paper is recognized for several positive aspects: (1) New benchmark with various irrational teachers and well-designed evaluation protocol, (2) benchmarking current SOTA preference-based RL algorithms, (3) detailed analysis on different algorithmic design choices, (4) clear write-up, and (5) well-documented (easy to install and run) codebase. We address your comments and questions below:
>
> ---
> **Q1. More environments**
>
> **A1.** To conduct an extensive study using various simulated teachers, we focused on hard environments like Quadruped, Sweep into, and Button Press. However, we also agree that adding more environments would be interesting to check whether we can get the same conclusion on different environments. We will include more environments in the final version of the benchmark.
>
> ---
> **Q2. Justifications on the irrationality model**
>
> **A2.**  We remark that the irrationalities we chose are largely motivated by the literature on preference learning: works often report the need to give people a third and even fourth option due to human inability to perfectly compare options (see for instance [Holladay et al., 2016], and we will add references to the paper as motivation). The Terry/Luce/Shephard model has been widely used in the literature on human choice since the 1950s, first introduced in mathematical psychology. The outlier here is myopia/recency effect bias, which we added as a hypothetical bias that is plausible for humans to have, mainly to add diversity to the benchmark. Of course, behavioral economics has identified hundreds of biases, and it is not the goal of the benchmark to cover all of them or to have an accurate human model, rather to stress-test the methods for reward learning with teachers that go beyond rational choice.
>
> Even though showing a relationship between the irrationality models and real human teaching (i.e. how much the proposed irrationality model is accurate) would be very valuable, the reality is that human behavior is nuanced and complex -- chances are that none of the biases in our simulated teachers perfectly correlate with real human behavior (rather, human behavior is explained by some combination of some subset of these with others that we do not yet model in the benchmark). The goal of the irrationalities is not to be a perfect model of human behavior, rather to enable evaluating the method with a diversity of possible teachers. The idea is that if a method can perform well with multiple teachers in the benchmark, then it has stronger chances of performing well with real teachers as well.

---

### Official Review · Reviewer_zxDS · 2021-07-06
**Interesting idea**

**Rating:** 7
**Confidence:** 2
**Clarity:** The writing is very clear and easy to…

**Strengths:**

1. The approach to simulate human input with a preference model is important and interesting.
2. The models used to validate the concept and serve as a reference are competitive (the current state-of-the-art approaches were used).
3. Accompanying code is documented sufficiently to reproduce the results and propose novel solutions.
4. Conducted experiments are valid and well-designed, e.g., the mean score averaged over several runs is reported. The paper and appendices describe them in detail, including hyperparameter description, analysis of learned reward functions, and learning curves.
5. The authors demonstrated the usefulness of the benchmark by using it to analyze algorithmic design choices.

**Weaknesses:**

1. It is questionable what is the relation between proposed simulated teachers and real human teachers. It would be very interesting to check what is the relation between proposed irrationalities and real human behavior. The authors can check this for tasks that require less agent experience from selected ones.
2. Since there are many possible irrationalities, there is a lack of explanation why authors choose only five of them (they are better than others in some way)?
3. Too much details about B-pref showcase, too less details about explaining crucial parts of the benchmark paper (see 1,2 points)


**Additional Feedback:**

1. Some of the tested irrationalities (human biases) are actually rational heuristics for modifying simulated teachers: “Skip”, “Equal”. The authors should divide them into groups with more intuitive names.
2. It would be interesting to summarize results for each irrationality separately (e.g. add extra rows with average numbers for all tasks for different irrationality and methods). Also, the table is very condensed. Maybe a heatmap would be more readable?
3.  Question to the authors: Results in Table 1 are not well discussed, e.g. for PrefPPO method the results on Equal, Skip and  Myopic irrationalities are sometimes better for smaller numbers of queries?
4. Small mistake: #288 “Ibarz et al. [30] improved this method the efficiency of this method by introducing additional” - repetition.
5. The expression $\sum_{t=1}^{H} \gamma_{my}^{H-t} r(s_{t}^{i}, a_{t}^{i}$ appears more than 10 times (including cases where $\gamma_{my}=1$), often inline. I suggest introducing a shorter notation, e.g. $R^i(\gamma_{my})$ and putting $R^i = R^i(1)$. This would improve readability. Also, the subscript "my" in $\gamma$ seems superfluous, as there is no other $\gamma$ in the paper.
6. At least a reference to the appendix should be given for information about some parameters, e.g. the values $\delta_{skip}$ and $\delta_{equal}$.


**Correctness:**

Conducted experiments are valid and well-designed, e.g., the mean score averaged over several runs is reported.

**Documentation:**

Accompanying code is documented sufficiently to reproduce the results and propose novel solutions. Moreover, the paper and appendices describe experiments in detail, including hyperparameter description, analysis of learned reward functions, and learning curves.

**Ethics:**

There are no ethical concerns related to this work.

**Relation To Prior Work:**

The reviewer has no expertise in RL, and therefore does not know prior work.

**Summary And Contributions:**

This paper proposes a benchmark for preference-based Reinforcement Learning with an evaluation of two key properties:
Performance of the RL agent under a fixed budget of feedback
Robustness to potential irrationalities (noise, mistake, skip, equal, myopic)
The key advantage of this benchmark is that we can very quickly compare preference-based RL methods by simulating a human teacher using common stochastic models in which one can manipulate its terms and operators to modify the importance of specific irrationality.
The authors demonstrated the usefulness of the benchmark by using it to analyze algorithmic design choices (e.g. selecting informative queries) for state-of-the-art methods: PEBBLE and PrefPPO.

---

> ### Author Response · Authors · 2021-07-10
> **Authors' Response**
>
> We sincerely thank you for your helpful feedback and insightful comments. We appreciate that our paper is recognized for several positive aspects: (1) New benchmark with various irrational teachers and well-designed evaluation protocol, (2) benchmarking current SOTA preference-based RL algorithms, (3) detailed analysis on different algorithmic design choices, (4) clear write-up, and (5) well-documented (easy to install and run) codebase. We address your comments and questions below:
>
> ---
> **Q1. Real human teacher**
>
> **A1.** We agree that showing a relationship between the irrationality models and real human teaching would be very valuable; however, the reality is that human behavior is nuanced and complex. Even though we picked representative biases (see A2), chances are that none of the biases in our simulated teachers perfectly correlate with real human behavior (rather, human behavior is explained by some combination of some aspects of these with others that we do not yet model in the benchmark). The goal of the irrationalities is not to be a perfect model of human behavior, rather to enable evaluating the method with a diversity of possible teachers. The idea is that if a method can perform well with multiple teachers in the benchmark, then it has stronger chances of performing well with real teachers as well. In the future, we plan to perform a follow-up user study that looks not so much at the correlation between the simulated teachers and real ones, but rather the correlation between the reward learning performance as evaluated by the benchmark, and the performance as evaluated with real humans. We believe that positive results there would serve to further strengthen the value of the benchmark.
>
> ---
> **Q2. Why 5 irrationalities?**
>
> **A2.** While we presented evaluations with only a few simulated teachers in our experiments, our benchmark parametrizes teachers in such a way that many irrationality levels and combinations could be used to produce a wide array of possible simulated teachers. That said, the reviewer is absolutely right to point out that we only focus on certain types or axes of irrationality. The ones we chose are largely motivated by the literature on preference learning: works often report the need to give people a third and even fourth option due to human inability to perfectly compare options (see for instance [Holladay et al., 2016], and we will add references to the paper as motivation). The Terry/Luce/Shephard model has been widely used in the literature on human choice since the 1950s, first introduced in mathematical psychology. The outlier here is the myopia/recency effect bias, which we added as a hypothetical bias that is plausible for humans to have, mainly to add diversity to the benchmark. Of course, behavioral economics has identified hundreds of biases, and it is not the goal of the benchmark to cover all of them or to have an accurate human model, rather to stress-test the methods for reward learning with teachers that go beyond rational choice.
>
> [Holladay et al., 2016] Holladay, R., Javdani, S., Dragan, A. and Srinivasa, S., Active comparison based learning incorporating user uncertainty and noise. In RSS Workshop on Model Learning for Human-Robot Communication, 2016.
>
> ---
> **Q3. Results in Table 1**
>
> **A3.** Thank you for your pointer. The performance of PrefPPO is relatively low compared to PEBBLE and it has high variances. Because of that, PrefPPO with small numbers of queries shows better performances in a few cases. To address this, we will run experiments with more random seeds and add related discussions in the final version.
>
> ---
> **Q4. Editorial comments/clarification**
>
> **A4.** Thank you very much for your detailed comments. We described how to set the values of $\delta_{skip}$ and $\delta_{equal}$ in Appendix D. We will incorporate the suggested editorial comments and clarifications in the final draft.

---

> > ### Comment · Reviewer_zxDS · 2021-07-20
> > **Response**
> >
> > Thank you for your comments, I'm changing my review from a 6 to a 7.

---

### Decision · Program_Chairs · 2021-07-26

**Decision:**

Accept

**Comment:**

The reviews for this paper are overwhelmingly positive. This paper proposes a very interesting challenge for learning policies using a teacher's preferences and pre-defined rewards which is called as preference-based RL.

The research problems that this paper investigates are important and crucial problem for reinforcement learning to deploy the RL agents in real-world problems. The benchmark and the paper focuses on RL agents under a fixed budget of feedback Robustness to potential irrationalities that may be present in many real world problems. The paper tests the algorithms with the two SOTA algorithms: PEBBLE and PrefPPO.

Some of the important feedback given by the reviewers and authors' response:
* Reviewer V29j commented that the the scale of the benchmarked environments in the paper is small and the authors responded that they will include more environments in the final version of the paper.
* Reviewer zxDS and Reviewer chBv using actual human feedback instead of the simulated feedback with irrationalities. The authors argued that although chances are that none of the biases in the simulated teachers perfectly correlate with real human behavior, it will be still useful for evaluating the method with a diversity of possible teachers.